# A biomimetic ocular prosthesis system: emulating autonomic pupil and corneal reflections

Seongchan Kim[1,2], Yoon Young Choi[3], Taewan Kim [4,5], Yong Min Kim [6], Dong Hae Ho[7], Young Jin Choi[8], Dong Gue Roe[9], Ju-Hee Lee[10], Joongpill Park[4], Ji-Woong Choi[11], Jeong Won Kim [11], Jin-Hong Park [1,10], Sae Byeok Jo [12], Hong Chul Moon [6] ✉, Sohee Jeong [4,5] ✉ & Jeong Ho Cho [8] ✉

The human light modulation response allows humans to perceive objects clearly by receiving the appropriate amount of light from the environment. This paper proposes a biomimetic ocular prosthesis system that mimics the human light modulation response capable of pupil and corneal reflections. First, photoinduced synaptic properties of the quantum dot embedded photonic synapse and its biosimilar signal transmission is confirmed. Subsequently, the pupillary light reflex is emulated by incorporating the quantum dot embedded photonic synapse, electrochromic device, and CMOS components. Moreover, a solenoid-based eyelid is connected to the pupillary light reflex system to emulate the corneal reflex. The proposed ocular prosthesis system represents a platform for biomimetic prosthesis that can accommodate an appropriate amount of stimulus by self-regulating the intensity of external stimuli.

Humans represent a complex of dozens of sophisticated organs that function to sustain life. In recent decades, artificial organs have emerged as promising replacements for human organs damaged by aging, injury, or disease[1–4]. With advancements in research on artificial organs, several researchers have attempted to implement perceptive prostheses based on a fundamental understanding of the human cognitive process to external stimuli[5–9]. This task involves emulating the phenomena that occur in the human body when subjected to external stimuli and converting these external stimuli into signals analogous to biological signals. A representative perceptive prosthesis that requires a biosimilar working mechanism is an artificial eye, which

provides vision, responsible for 80% of the information perception by humans[10–14]. Although recent approaches associated with artificial eyes have achieved satisfactory recognition of external light, research on mimicking human ocular reflexes according to the intensity of external light is limited. Human ocular reflexes can be divided into pupillary light reflex and corneal reflex. The former reflex is realized by changing the diameter of the pupil depending on the intensity of incident light, similar to the aperture of cameras. The latter reflex aims at blocking a certain amount of incident light through "eye blinking". Both behaviors are governed by synaptic weight updates induced by a light stimulus. Therefore, the adoption of photonic synapses with bio-

[1]SKKU Advanced Institute of Nanotechnology (SAINT), Sungkyunkwan University, Suwon 16419, Korea. [2]Department of Engineering Science and Mechanics, The Pennsylvania State University, University Park, PA 16802, USA. [3]Department of Mechanical Science and Engineering, University of Illinois at Urbana Champaign, Urbana, IL 61801, USA. [4]Department of Energy Science and Center for Artificial Atoms, Sungkyunkwan University, Suwon 16419, Korea. [5]Sungkyun Institute of Energy Science and Technology (SIEST), Sungkyunkwan University, Suwon 16419, Korea. [6]Department of Chemical Engineering, University of Seoul, Seoul 02504, Korea. [7]Mechanical Engineering, Soft Materials and Structures Lab, Virginia Tech, Blacksburg, VA 24061, USA. [8]Department of Chemical and Biomolecular Engineering, Yonsei University, Seoul 03722, Korea. [9]School of Electrical and Electronic Engineering, Yonsei University, Seoul 03722, Korea. [10]Department of Electrical and Computer Engineering, Sungkyunkwan University, Suwon 16419, Korea. [11]Korea Research Institute of Standards and Science (KRISS), Daejeon 34113, Korea. [12]School of Chemical Engineering, Sungkyunkwan University (SKKU), Suwon 16419, Korea. ✉e-mail: hcmoon@uos.ac.kr; s.jeong@skku.edu; jhcho94@yonsei.ac.kr

similarity that can realize analog signal transmission is of significance in realizing artificial illuminance modulation responses.

A notable feature of photonic synapses is that the synaptic weights are adjusted by external light stimuli, unlike conventional artificial synapses, in which the synaptic weights are controlled by electrical signals[13–17]. Because a photonic synapse can realize both light sensing and signal processing (that is, can function as both a photosensor and artificial synapse), the use of such a synapse scan decreases the circuit complexity and increases the integration density when constructing artificial eyes[16]. Various materials such as organics[18,19], two-dimensional materials[20,21], and perovskite[22,23] have been recommended for the fabrication of photonic synapses. However, the low reliability, inferior stability, and small-area processing have hindered the practical application of photonic synapse-based prostheses. These issues can be alleviated by introducing highly stable metal oxide semiconductors in integrated circuits[24–28]. Moreover, the use of metal oxide semiconductors is advantageous in that the photoinduced generation of metastable oxygen vacancies in the metal oxide semiconductors induces persistent photocurrent (PPC) characteristics, leading to long-term potentiation (LTP) characteristics under pulsed light stimuli[25–28]. Despite the excellent properties of photonic synapses, the implementation of long-term depression (LTD) (i.e., ability to return to the original state) and LTP characteristics under a continuous light input signal rather than a pulsed light signal, as in the human visual perception, remains a challenge.

Considering these aspects, this study was aimed at developing a biomimetic ocular prosthesis system that can mimic the human light modulation response by using a quantum dot (QD) embedded photonic synapse (QEPS) that uses biosimilar signals. The proposed system was composed of a solenoid-based artificial eyelid (s-eyelid), an electrochromic (EC) device, and QEPS-based neuron circuits, corresponding to the biological eyelid, pupil, and visual system (retina and optic nerve), respectively. Input light was efficiently blocked and attenuated through the s-eyelid and EC devices, respectively, thereby filtering the excess light. The working principle of the prosthetic components was based on the synaptic properties of QEPS: By embedding tetrahedron-shaped indium phosphide (InP) QDs in indium gallium zinc oxide (IGZO), LTP and LTD characteristics were successfully achieved in the QEPS. Finally, the signal transmission for light intensity modulation reflexes was successfully demonstrated with an ocular prosthesis system capable of both corneal reflex and pupillary light reflex. The proposed ocular prosthesis system with biosimilar operating characteristics can provide a platform for prostheses that can suitably respond to external stimuli.

## Results

### A biomimetic ocular prosthesis consisting of pupillary light reflex and corneal reflex

Humans detect light stimuli through photoreceptors in the retina to cognize visual information. The light stimulus, which is converted into an electrical signal, and the inhibitory signal, which controls the synaptic weight to light, constitute a postsynaptic signal at the synapse and concretize the visual information (Fig. 1a). To clearly perceive objects, it is necessary to receive an appropriate amount of light from the environment. The regulation of incident light illuminance is achieved through the pupillary light reflex and corneal reflex, as shown in Fig. 1b. In the former case, the intensity of light is controlled by the movement of the pupil: Dilation of the pupil allows light with a larger intensity to reach the retina, whereas contraction of the pupil limits the intensity of incident light. When humans are exposed to an unadaptable intensity of light, light penetration is blocked by simply closing the eyelids (i.e., blinking). These reflex responses can be simulated using an ocular prosthesis system consisting of the s-eyelid, EC device, and QEPS, as shown in Fig. 1c. In this system, the QEPS receives external light stimuli and generates a postsynaptic current (PSC). The PSC is

delivered to both the EC device and s-eyelid, corresponding to the pupil and eyelid in the biological reflex system, respectively. Each electric component must exhibit desirable and distinct properties in the operation. For the QEPS, the PSC must exhibit LTP/D characteristics under a continuous light input (notably, the LTP/D characteristics of conventional photonic synapses have been initiated to stimulus in the form of pulses) to provide a reflex against an actual surrounding condition. In other words, the QEPS must present LTP/D characteristics under continuous illumination and dark conditions, respectively, when an inhibitory voltage ($V_{Inh}$) pulse train is applied, similar to the human ocular system. In particular, the application of a continuous biosimilar $V_{Inh}$ pulse train eliminates the need to isolate the weight control terminals, thereby simplifying the circuit design. This photoresponse of the QEPS arises from the PPC property associated with the photogenerated metastable oxygen vacancy traps in IGZO (LTP) and the role of InP QDs as an electron releaser (LTD). Supplementary Fig. 1 compares the LTP/D characteristics of IGZO-only photonic synapse (IPS) and QEPS, with complete recovery of PSC to the initial state observed only in the QEPS. Furthermore, for the EC device, light illuminance modulation is realized by the coloration of the device upon the application of voltage, yielding a decrease in transmittance. To this end, EC gels are prepared by introducing diheptyl viologen ($DHV^{2+}$, cathodic EC material) and dimethyl ferrocene (dmFc, counter anodic material) into a deformable ion gel consisting of polystyrene-*ran*-poly(methyl methacrylate) (PS-*r*-PMMA) and 1-ethyl-3-methylimidazolium bis(trifluoromethyls ulfonyl) imide ([EMIM][TFSI])[29]. The resulting gel exhibits EC behaviors. For example, when an external voltage is applied, $DHV^{+\cdot}$ is produced, and the gel turns blue[30]. The s-eyelid controls the size of the hole by which light travels, through the voltage-induced linear actuation of the solenoid.

Figure 1d shows a schematic of the signal flow of the biomimetic ocular prosthesis system. The external light input first passes through the s-eyelid followed by the EC device. Next, the filtered light induces the QEPS to generate the PSC which is subsequently delivered to the EC device and neuron circuit that modulates the light intensity. When the incident light power is within the adaptable range, the increase in PSC lowers the transmittance of the EC device (pupillary light reflex, left panel of Fig. 1e), suppressing any further increase in the PSC. A stronger incident light intensity with unacceptable optical power triggers an actuation voltage ($V_{Act}$) in the neuron circuits (corneal reflex, right panel of Fig. 1e). $V_{Act}$ is applied to the s-eyelid to screen input light and induces LTD in the QEPS.

### Comparison of electrical properties of photonic synapses

Figure 2a shows a cross-sectional schematic of the QEPS with the tetrahedron InP QDs[31] embedded in the IGZO. For the preparation, the InP QDs were coated onto a 100-nm-thick $SiO_2$/highly doped Si wafer, followed by ligand exchange with ammonium thiocyanate ($NH_4SCN$). Next, the IGZO precursor solution was spin-coated onto the InP QDs to form the embedded structure. Subsequently, the QD-embedded IGZO (photoactive layer) was patterned via conventional photolithography and chemical etching. Finally, the Al electrode was deposited on the photoactive layer to form drain and source electrodes corresponding to presynaptic and postsynaptic terminals, respectively. Highly doped Si was used to prepare the gate electrode (weight control terminal) to which $V_{Inh}$ was applied. As the light source, a 520 nm laser was used to measure the light dependence of the PSC. Figure 2b shows the current–voltage characteristics (PSC versus presynaptic voltage ($V_{Pre}$) and drain current versus drain voltage) of the IPS and QEPS under various light intensities (dark to 10 µW) at $V_{Inh} = 0$ V (Supplementary Fig. 2 shows the optoelectronic performance). In the case of the QEPS, a large variation in the PSC can be observed ($1.47 \times 10^{-5}$ A, light intensity = 10 µW, $V_{Pre} = 5$ V). The corresponding value for the IPS is $7.75 \times 10^{-6}$ A under illumination (light intensity = 10 µW, $V_{Pre} = 5$ V). To investigate the mechanisms that induce the large modulation of PSC in

the QEPS, the distribution of the trap states affecting the charge transport behavior was extracted from the current–voltage plot in the dark condition (Fig. 2c). In the logarithmic scale (Supplementary Fig. 3), three regions can be distinguished, as indicated by straight lines, as shown in. Region $X$ corresponds to Ohmic behavior, and region $Z$ follows Mott's trap-free $V^2$ law[32,33]. Region $Y$ corresponds to the trap-limited space charge limited current, and the current follows the Mark–Helfrich equation[33–35]:

$$I = Aq\mu N_L \left(\frac{\varepsilon_r \varepsilon_0}{q N_{t,0} e^{-l}}\right)^l \left(\frac{l}{l+1}\right)^l \left(\frac{2l+1}{l+1}\right)^{l+1} \frac{V^{l+1}}{L^{2l+1}} \tag{1}$$

where $A$ indicates the channel area, $q$ is the elementary charge, $L$ is the channel length, and $\varepsilon_r$ and $\varepsilon_0$ represent the relative permittivity and permittivity of vacuum, respectively. The total trap density below the

mobility edge ($N_{t,0}$) can be expressed as $N_{t,0} = N_L k T_c$, where $T_c$ represents the characteristic temperature of the trap, $k$ is the Boltzmann constant, and $N_L$ is the number of sites on the lowest unoccupied molecular orbital. $T_c$ was estimated using the relation $T_c = lT$, where $T$ is the temperature during the current–voltage measurement. Through linear fitting in Region $Y$, $l$ was calculated to be 3.30 and 2.14, indicating that the $T_c$ values in the IPS and QEPS are 999.13 K (0.085 eV) and 643.3 K (0.055 eV), respectively. Here, the energy distribution was calculated using the relation $E_c = kT_c$, where $E_c$ represents the thermal energy at a characteristic temperature. Furthermore, the trap densities of IPS and QEPS were calculated to be $2.18 \times 10^{17}$ and $1.60 \times 10^{16}$ cm$^{-3}$, respectively, using the Mark–Helfrich equation. Figure 2d shows the schematic band diagram of the model. The QEPS exhibits a shallow trap density-of-states (DOS) distribution compared to the IPS, despite the low conductivity of InP QDs in the IGZO layer. In addition to the lower trap density, this trap distribution supports that the movement

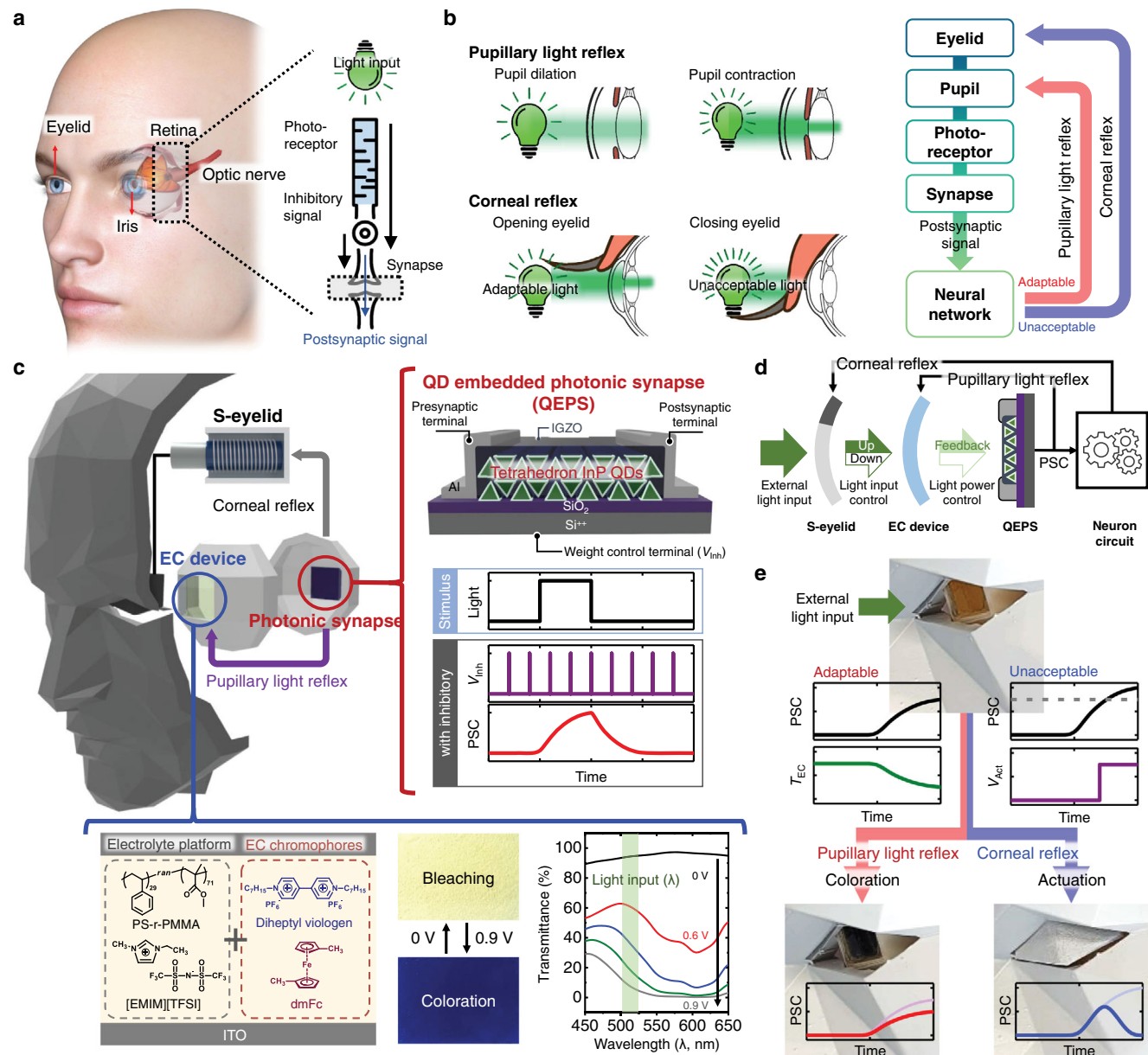

Fig. 1 | Schematic of a biomimetic ocular prosthesis consisting of pupillary light reflex and corneal reflex. Schematic of a concretization of the visual information; b regulation of incident light illuminance in a biological system; c ocular prosthesis

system. Schematic of signal flow in d ocular prosthesis system, (left panel of e) pupillary light reflex, and (right panel of e) corneal reflex.

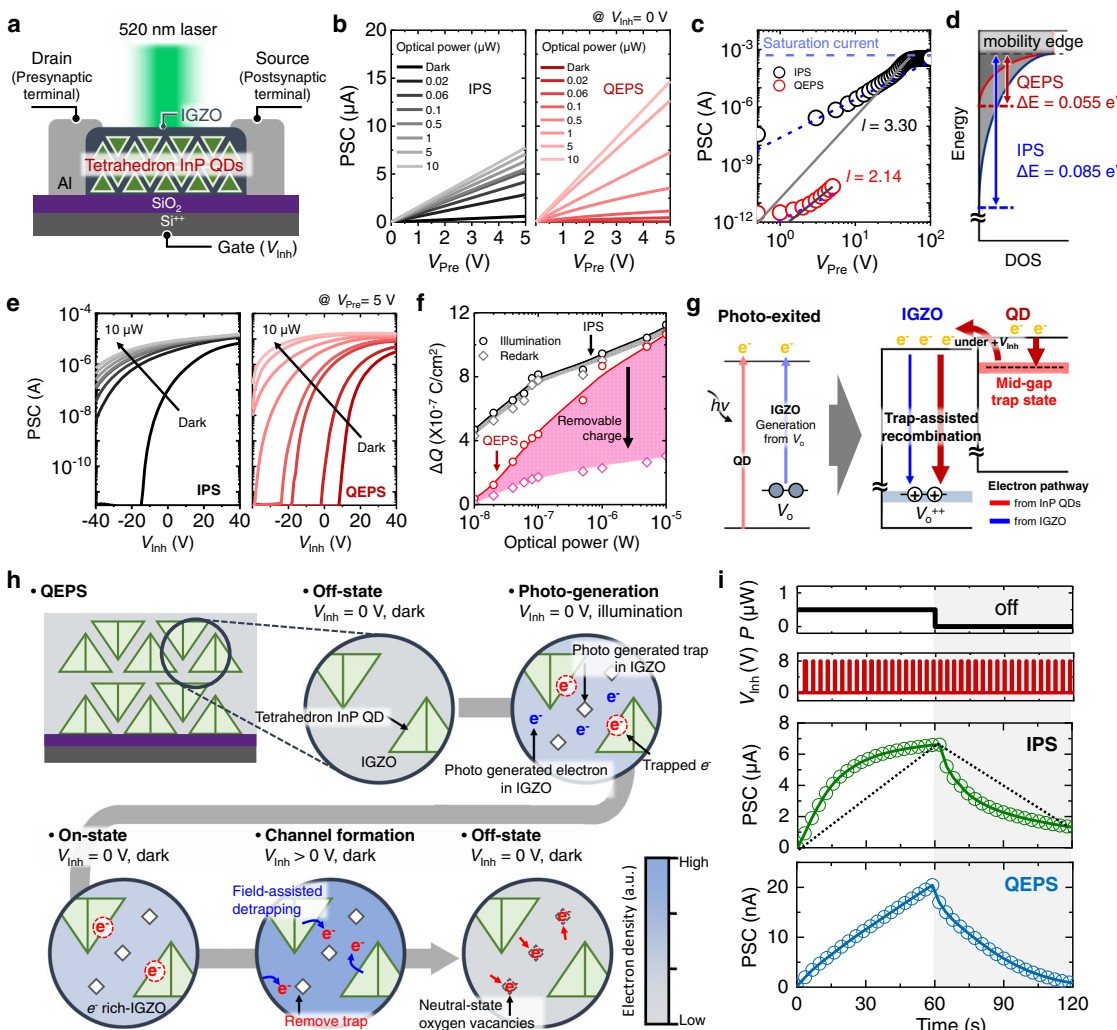

**Fig. 2 | Comparison of electrical properties of photonic synapses. a** Cross-sectional schematic of the QEPS. **b** Current–voltage characteristics of the IPS (left panel) and QEPS (right panel). **c** Logarithmic plot of current-voltage curves of photonic synapses in dark conditions ($V_{Inh} = 0$ V). **d** Energy band diagram of the trap distribution in photonic synapses. **e** Semilogarithmic plot of transfer curves of the IPS (left panel) and QEPS (right panel). **f** Plot of charge difference versus optical power. The difference between the illumination (circle) and re-dark (diamond) states represents the removable charge. Schematic of **g** electron pathway and **h** recombination process in the QEPS. **i** LTP/D behavior of IPS (green) and QEPS (blue).

of carriers in the QEPS is less disturbed by traps present in the channel[35–37] (see also Supplementary Fig. 4), by which the higher PSC of QEPS under illumination can be explained.

Figure 2e shows the PSC modulation during sweeping of the $V_{Inh}$ (i.e., transfer characteristics) from −40 to 40 V under various light conditions (dark to 10 μW). The threshold voltage ($V_{Th}$) shift increases with increasing light intensity, resulting in the PSC modulation of each photonic synapse. The influence of $V_{Inh}$ on the PSC was determined from the transfer characteristics measured under the dark–illumination–dark (re-dark) condition, as shown in Supplementary Fig. 5. In general, the charge difference ($\Delta Q$) between the original dark state and conditioned states (illumination state and re-dark state) can be expressed as $\Delta Q = C\Delta V_{Th}$, where $C$ indicates the SiO₂ capacitance (see Supplementary Fig. 6 for $\Delta V_{Th}$). $\Delta Q$ represents the carriers that contribute to the increase in the PSC of each photonic synapse. Moreover, the difference in $\Delta Q$ between the illumination and re-dark state indicates the charges that can be removed from the IGZO channel by applying $V_{Inh}$, as shown in Fig. 2f. In the case of the QEPS, a large difference in $\Delta Q$ is observed between the illumination and re-dark states, whereas there is no notable difference for the IPS. Here, we define the difference in $\Delta Q$ as a removable charge which means a photogenerated carrier that can be easily removed. Note that the

hysteresis behavior in Supplementary Fig. 6 originates from the removable charge. The removable charge in QEPS is a result of the mid-gap trap originating from the P dangling bond in the NH₄SCN passivated InP QDs, as evidenced by the trap emission in the wavelength region above 800 nm[38,39], as shown in Supplementary Fig. 7. Photogenerated electrons trapped in the mid-gap state on the surface of the InP QDs escape to IGZO when $V_{Inh}$ (gate voltage) is applied, causing electron–hole recombination in IGZO (Fig. 2g). To characterize the mid-gap trap state, ultraviolet photoelectron spectroscopy (UPS) and inverse photoemission spectroscopy (IPES) of InP QDs was further investigated (Supplementary Fig. 8). The unoccupied DOSs near the conduction band edge exhibit clear tail states compare to the occupied DOSs near the valence band edge. The position of distribution near the conduction band edge is located broadly at 0.5–1.49 eV above the fermi level ($E_f$). The tail states are associated with the trap states near the conduction band edge and the PL emission above 800 nm. The energy difference between the mid-gap trap and conduction band edge (-0.8 eV) is reduced due to the localized tail states (-0.25 eV) which facilitates the filed-assisted detrapping to IGZO. Figure 2h illustrates the recombination process in the QEPS (upper part of QEPS) more in detail. Under illumination (2nd circle, photogeneration), the neutral-state oxygen vacancies are ionized, resulting in PPC

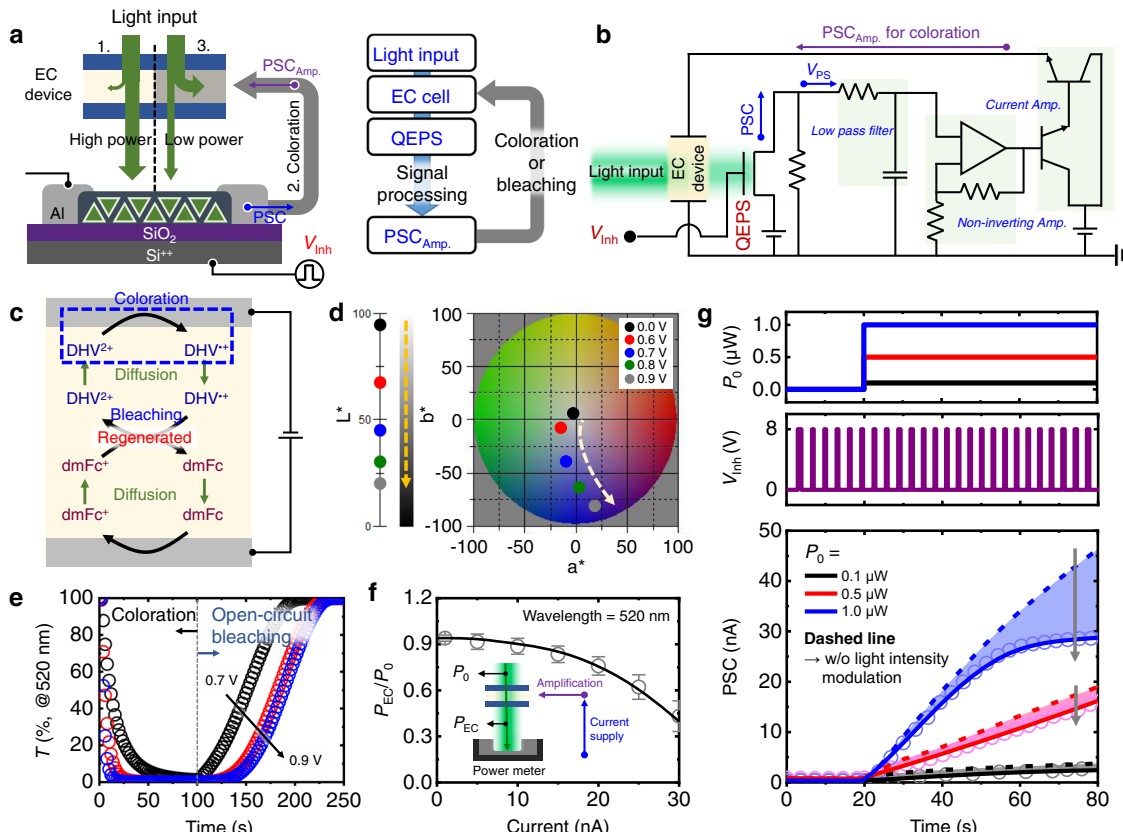

**Fig. 3 | Artificial pupillary light reflex system with EC device. a** Schematic of artificial pupillary light reflex (left panel) and signal flow (right panel). **b** Circuit diagram of artificial pupillary light reflex. **c** Schematic of electrochemical reactions in the EC device. **d** CIELab color coordinates ($L^*$, $a^*$, $b^*$) of the EC device during coloration. **e** Transient transmittance of the EC device at 520 nm. **f** Plot of $P_{EC}/P_0$ versus current, showing the light intensity modulation capability of the EC device. Error bars represent standard deviation ($n = 4$) **g** Light modulation properties under the application of light input and $V_{Inh}$, resulting in PSC modulation.

characteristics ($V_O \rightarrow V_O^{2+} + 2e^-$; $V_O^{2+}$ and $e^-$ denote photogenerated traps and electrons, respectively). Moreover, at the InP QDs, the photogenerated electrons are trapped in the mid-gap trap on the surface, as mentioned previously. The increased electron concentration and electrons trapped by the InP QDs remain even after the light is removed due to the PPC characteristics (3rd circle, on-state) and trapping, respectively. When a positive $V_{Inh}$ is applied (4th circle, channel formation) in a continued dark state, the electrons trapped in the mid-gap trap of InP QDs escape to IGZO to form a channel between the InP QDs. The increase in the electron concentration of IGZO facilitates the reverse reaction to eliminate the ionized oxygen vacancies ($V_O^{2+} + 2e^- \rightarrow V_O$). Consequently, the $V_O^{2+}$ concentration decreases and the QEPS channel returns to its original state (5th circle, off-state). In brief, the mid-gap trap of InP QDs causes a removable charge that can easily return to its original state after applying $V_{Inh}$ under off-state. This electron trapping–detrapping cycle occurs in the entire QEPS channel, as shown in Supplementary Fig. 9, as the InP QDs are distributed throughout the IGZO, leading to a large amount of removable charge. In contrast, in the case of the IPS, the amount of removable charges contributing to the PPC characteristics is limited as the cycle occurs only near the gate insulator (Supplementary Fig. 10). These PPC characteristics and the removable charge can be utilized for the synaptic weight adjustment of the photonic synapse (Supplementary Figs. 11 and 12 show the short-term behavior of QEPS). Figure 2i shows the LTP/D behavior of the IPS and QEPS (also see Supplementary Figs. 13 and 14). Both photonic synapses were exposed to 520 nm of light with an intensity of 0.5 μW for 60 s and placed in dark conditions for 60 s while continuously applying the $V_{Inh}$ pulse train. Both photonic synapses exhibit LTP characteristics in response to the photo stimulus. However, the photonic synapses exhibit

different LTD behaviors: In the case of the IPS, the PSC does not return to its original state due to the insufficient amount of removable charge. In contrast, the presence of abundant removable charge allows the increased PSC of the QEPS to return to the initial level during the multiple cycles of illumination and dark conditions (Supplementary Fig. 15). Note that the different recombination processes in the IPS and QEPS originate from the absence or presence of mid-gap traps in InP QDs. To verify the role of the mid-gap trap in QEPS, we further investigate the mid-gap trap engineered QEPS through HF treatment[40,41] (Supplementary Fig. 16). The result exhibits a similar tendency in the excitatory postsynaptic current (EPSC) whereas comparable change was observed in the inhibitory postsynaptic current (IPSC) (Supplementary Figs. 17 and 18). As a result of the different IPSC behavior exhibited by the increase in the mid-gap trap, sufficient removable charge facilitates returning the increased PSC to its original state, as shown in Supplementary Fig. 19. Based on these characteristics the QEPS with sufficient removable charge exhibits linear LTP/D characteristics owing to the suppressed trap generation by the continuous $V_{Inh}$ pulse train (Supplementary Fig. 20). To further validate the uniformity of the QEPS, LTP/D characteristics were measured on a 6 × 6 array of QEPS as shown in Supplementary Fig. 21. The training/recognition task of Modified National Institute of Standards and Technology (MNIST) digit patterns was performed using the 36 devices. In this simulation, the array of QEPS shows an average recognition accuracy of 91.7% (Supplementary Fig. 22).

## Artificial pupillary light reflex system with EC device

Figure 3a schematically illustrates the artificial pupillary light reflex system. A continuous light stimulus is converted by the QEPS to an increase in the PSC, leading to the coloration of the EC

device. The decrease in transmittance of the EC device and application of the repetitive $V_{Inh}$ pulse train decreases the PSC. This synaptic behavior of the device is controlled by the circuit of the artificial pupillary light reflex system, which includes an EC device, the QEPS, and CMOS components (Fig. 3b). A low-pass filter and amplifiers (voltage and current amplifier) are used for signal filtering and amplification, respectively. Figure 3c depicts the overall electrochemical reactions related to the coloration and bleaching of the EC device. When the $DHV^{2+}$ concentration decreases near the cathode due to high PSC, blue $DHV^{+,\cdot}$ is produced. Moreover, dmFc is oxidized near the counter anode, thereby completing the circuit. As coloration proceeds, the blue color becomes intense and reaches the CIELab color coordinates ($L^*$, $a^*$, $b^*$) of (19.07, 17.28, −71.61) (Fig. 3d). In general, a decrease in $L^*$ (95–19.07) implies a decrease in the transmittance of the EC device. When the PSC level is low, the voltage supply to the EC device is cut off. Consequently, the reduced/oxidized ions diffuse from the electrode toward the bulk gel due to the concentration gradient. Eventually, the concentration profiles of $DHV^{+,\cdot}$ and $dmFc^+$ overlap and return to the original state (bleaching). To evaluate the dynamic responsiveness, the transient transmittance ($T$) of the EC device at 520 nm was measured (Fig. 3e). As the applied voltage increases, the EC device exhibits a faster coloration response, larger transmittance contrast, and higher coloration efficiency (Supplementary Table 1 and Supplementary Fig. 23). Moreover, the device colored by 0.9 V exhibits the longest bleaching time in the open-circuit condition, indicating the highest memory effect, that is, the ability to maintain the colored state in the absence of a power source. The delay in coloration and bleaching is advantageous in terms of biosimilar behavior (adaptation).

The effective modulation of light intensity by the EC device is shown in Fig. 3f and Supplementary Fig. 24. To ensure that the experimental conditions were equivalent to those of the overall system, a laser beam (520 nm of wavelength) was passed through the EC device ($P_{EC}$) while supplying a current of 1 nA, similar to the initial current level of the PSC. As the amount of current supply increases, the attenuation of the power is more significant due to the stronger coloration of the EC device, resulting in lower $P_{EC}/P_0$ ($P_0$ represents the input light intensity). Subsequently, $P_0$ and $V_{Inh}$ are applied to the QEPS to measure the PSC controlled by the artificial pupillary light reflex, as shown in Fig. 3g. When the intensity of the $P_{EC}$ is insufficient for the reflex ($P_0 = 0.1$ and $0.5\,\mu W$), the difference in the PSC is not significant for any light modulation. However, when a stronger light is input, the decrease in the PSC is pronounced due to the considerably reduced $P_{EC}$ through the EC device. The difference in the PSC with pupillary light reflex ($PSC_{reflex}$) and that without light intensity modulation ($PSC_{w/o}$) according to the optical power of the input light is shown in Supplementary Fig. 25. The large PSC difference at a higher optical power indicates that the artificial pupillary light reflex system can effectively adjust the PSC based on the feedback process between the QEPS and EC device.

### Biomimetic ocular prosthesis system capable of corneal reflex and pupillary light reflex

A biomimetic ocular prosthesis system capable of both corneal reflex and pupillary light reflex was constructed by incorporating the corneal reflex system and pupillary light reflex system. Figure 4a, b illustrate the overall schematic and corresponding circuit diagram of the ocular prosthesis system, respectively. The EC device and QEPS are attached to the 3D-printed sclera to serve as the pupillary light reflex system. To prepare the artificial corneal reflex system, the 3D-printed eyelid is connected to a solenoid (Supplementary Fig. 26) and assembled with the 3D-printed face. Furthermore, to operate the s-eyelid, a neuron circuit is designed such that the postsynaptic voltage ($V_{PS}$) enters a

Schmitt trigger. In particular, the Schmitt trigger emits a constant output ($V_{Act} = −6\,V$) to close the s-eyelid when the input amplitude exceeds the upper threshold level ($V_{th} = 3\,V$) and leads to immediate restoration when the increased input amplitude decreases below the lower $V_{th}$ of 1 V ($V_{Act} = +10\,V$). Figure 4c shows photographs of the pupillary light reflex and corneal reflex in the light-modulating ocular prosthesis system. Steps 1−3 depict pupillary light reflex when the synaptic weight of the QEPS is implemented under adaptive light illumination. Steps 3 and 4 depict the corneal reflex when the system is exposed to unacceptable light. Figure 4d shows the signals that govern the two reflexes. First, when the s-eyelid is in the open state, the EC device in the bleached state transmits the input light (520 nm of wavelength) with a slight decrease in power. Subsequently, $V_{PS}$ increases due to the increased synaptic weight in the QEPS. As the $V_{PS}$ increases (steps 1−3), $PSC_{Amp}$ increases, resulting in a decrease in $P_{EC}$ (pupillary light reflex). As the $V_{PS}$ exceeds the upper $V_{th}$ (step 4), the voltage for closing the s-eyelid ($V_{Act}$) is generated. Moreover, the bleaching of the EC device occurs due to the LTD properties of the QEPS in the dark condition and applied $V_{Inh}$ pulse train (light-gray shaded region in Fig. 4d). $V_{Act}$ is restored by decreasing the synaptic weight as $V_{PS}$ reaches the lower $V_{th}$ (opening s-eyelid, steps 4 to 2). Finally, after the eyelid opening, the $V_{PS}$ is controlled according to the light input intensity. This signal-processing demonstration suggests that the proposed light-modulating ocular prosthesis system can successfully emulate the human ocular system with biosimilar signals.

## Discussion

We established a biomimetic ocular prosthesis system by incorporating an s-eyelid, an EC device, a QEPS, and CMOS components. The proposed system could successfully emulate the human pupillary light reflex and corneal reflex under a continuous light input and biosimilar repetitive inhibitory voltage train. Signal transmission for biosimilar signal-mediated reflexes was performed via the QEPS, in which the LTP/D properties were successfully controlled via IGZO channels with tetrahedral InP QDs. Based on the synaptic properties of the QEPS, the light illuminance was effectively modulated by the complementary combination of the artificial pupillary light reflex system and artificial corneal reflex system, allowing modulation in both acceptable and unacceptable light intensity conditions. The proposed biomimetic prosthesis system and its underlying principle of organically responding to various light intensities can promote research on biomimetic prostheses.

## Methods

### Material preparation

To prepare the InP QD, 0.001 M tris(dimethylamino)phosphine (($Me_2N)_3P$) solution dissolved in oleylamine (OLA) was rapidly injected into a 0.001 M indium trichloride ($InCl_3$) solution dissolved in OLA at 250 °C and maintained for 1 h[31]. The synthesized InP QD was washed using butanol and redispersed in octane (50 mg/ml). To prepare the ligand exchange, 0.064 M ammonium thiocyanate ($NH_4SCN$) was dissolved in acetonitrile and stirred at room temperature for 12 h. The IGZO precursor solution was prepared by dissolving 0.085 M indium nitrate hydrate, 0.0125 M gallium nitrate hydrate, and 0.0275 M zinc acetate dehydrate in 2-methoxyethanol. The IGZO precursor solution was stirred at 75 °C for 12 h. To obtain the EC gel solution, diheptyl viologen was prepared by exchanging halogen anions ($Br^-$) with hexafluorophosphate ($PF_6^-$) in 1,1'-diheptyl-4,4'-bipyridinium di-bromide ($DHV(Br)_2$) to increase the solubility. The ionic liquid, 1-ethyl-3-methylimidazolium bis(trifluoromethylsulfonyl)imide ([EMI][TFSI]), was prepared via an anion exchange reaction between 1-ethyl-3-methylimidazolium bromide ([EMI][$Br^-$]) and excess lithium bis(trifluoromethyl-sulfonyl)imide (LiTFSI) in deionized (DI) water. The prepared diheptyl viologen and ionic liquid were dissolved in acetone (the amount of EC redox materials was 0.1 M). Furthermore, the

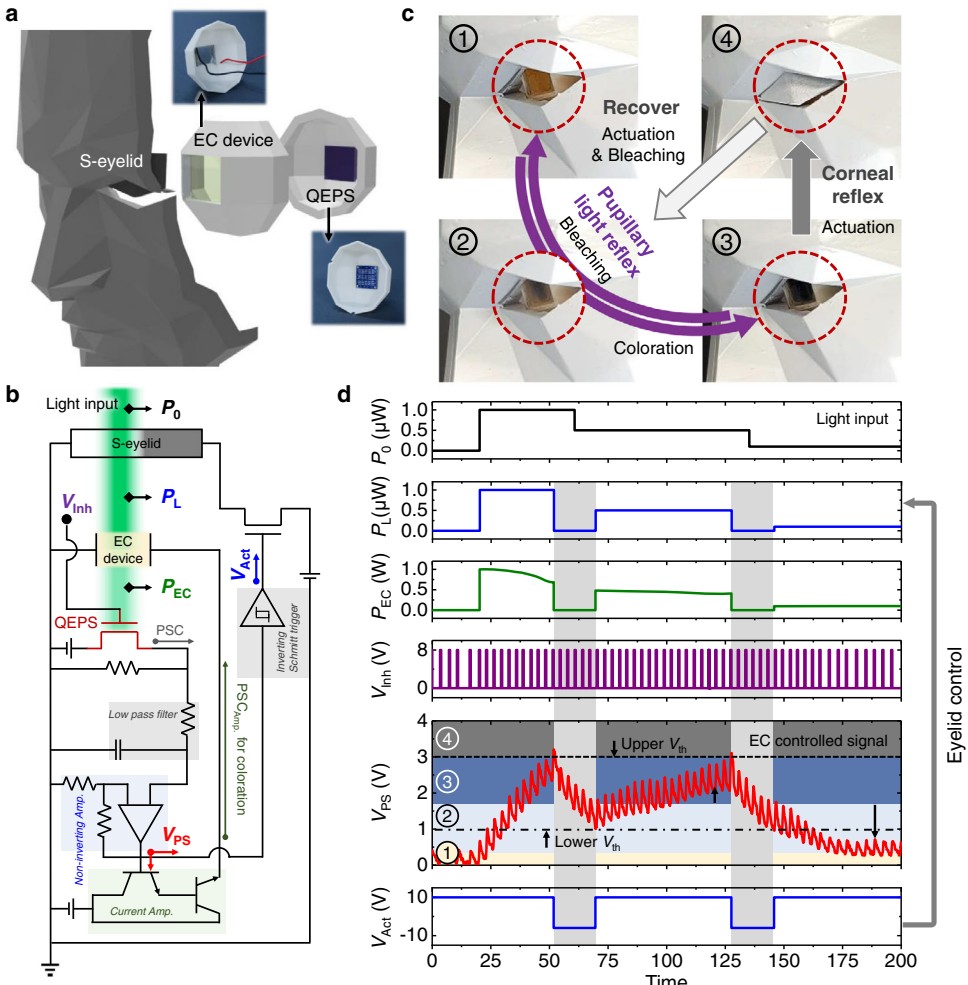

**Fig. 4 | Biomimetic ocular prosthesis system capable of corneal reflex and pupillary light reflex. a** Schematic and **b** circuit diagram of the ocular prosthesis system. **c** Real-time photographic images of pupillary light reflex and corneal reflex. **d** Signals in the process of self-regulating the intensity of external stimuli.

copolymer gelator, polystyrene-*ran*-poly(methyl methacrylate) (PS-*r*-PMMA), was prepared via reversible addition-fragmentation chain transfer polymerization. Moreover, 30 wt% of the copolymer gelator was mixed with the prepared solution.

## Device fabrication

To obtain the QEPS, the prepared InP QD solution was spin-coated at 1000 rpm for 180 s onto a Si wafer substrate with a 100-nm-thick $SiO_2$ layer, followed by ligand exchange using the $NH_4SCN$ solution. The InP QD film was dried at 100 °C for 2 min. The prepared IGZO solution was spin-coated at 4000 rpm for 30 s onto the substrate and dried at 60 °C for 1 min. The QD-embedded IGZO layer was sintered at 300 °C for 2 h in ambient conditions. To form the channel, the layer was patterned through conventional photolithography (AZ 5214E) and chemical etching (3 vol% LCE-12 (Cyantek Co.), dissolved in distilled water). To form the drain and source electrodes, Al patterns with a thickness of 30 nm were deposited by thermal evaporation through a shadow mask. To prepare the IPS, all the processes were identical except for the deposition of the InP QD layer. To obtain the EC device, the prepared EC gel solution was drop-cast onto the ITO-coated glass and dried in ambient conditions for 1 h. The EC device was fabricated by assembling ITO-coated glass, with double-sided adhesive tapes placed as spacers.

## System design

For the artificial pupillary light reflex system, a 10 MΩ resistor was used as the voltage divider. The low-pass filter was composed of a resistor (10 MΩ) and capacitor (470 nF). For the non-inverting amplifier, the signal was amplified by a factor of 31 through a comparator (OP07DP, Texas Instruments) and resistors (300 and 10 MΩ). Finally, the current flowing through the EC device was amplified through a current amplifier composed of two bipolar junction transistors (ZTX851STZ, Diodes Incorporated). For the artificial corneal reflex system, an inverting Schmitt triggers circuit was designed using a comparator (UA741CP, Texas Instruments) and resistors (1 and 9 kΩ). To operate the system, a solenoid (LY-01-LE DC 12 V, Toolparts) was connected to a transistor (R6030KNX, ROHM Semiconductor) driven by $V_{lid}$.

## Characterization

The absorbance and PL properties of the InP QDs were measured using the UV3600 (Shimadzu) and FLS1000 (infrared detector installed, Edinburgh) devices under ambient conditions, respectively. Photoelectron spectroscopy was performed using a hemispherical electron energy analyser (PSP Resolve120) and a He I discharge source ($\hbar\omega$ = 21.2) for UPS with an energy resolution of 0.15 eV. IPES was performed using a homemade IPES system comprising a BaO-cathode electron gun and a photon detector with a KBr-coated microchannel plate and $BaF_2$ window, which has a total energy resolution of 0.6 eV. The electrical properties of the QEPS were measured using a Keithley 4200A-SCS unit (Tektronix) and a 520 nm laser (MLL-III-520L, CNILaser) at room temperature under vacuum conditions. To evaluate the EC device performance, the changes in the optical property upon the application of external voltages were recorded on a UV–vis

spectrometer (V-730, Jasco) with a scan rate of 400 nm/min. DC voltage was supplied from a source meter (Keithley 2400, Tektronix). To investigate several characteristics of the devices, including the CIELab color coordinates, dynamics, and coloration efficiency, real-time variations in the transmittance and current were monitored using the same UV–vis spectrometer and a potentiostat (Wave Driver 10, Pine Instrument).

## Data availability

The data that support the plots within this paper and other findings of this study are available from the corresponding author upon reasonable request.

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

## Acknowledgements

J.H.C. was supported by Creative Materials Discovery Program through the National Research Foundation of Korea (NRF) funded by the Ministry of Science and ICT (NRF-2019M3D1A1078299), the Basic Science Program (NRF-2020R1A2C2007819) through the National Research Foundation (NRF) of Korea funded by the Ministry of Science and ICT, Korea, and the Technology Innovation Program (or Industrial Strategic

Technology Development Program) (20021909, Development of $H_2$ gas detection films (≤0.1%) and process technologies) funded by the Ministry of Trade, Industry & Energy (MOTIE, Korea).

## Author contributions

H.C.M., S.J., J.H.C. conceptualized and supervised this work. S.K. designed the experiments and carried out most of the experimental work and data analyses. T.K. and J.P. synthesized and analyzed the QDs. Y.M.K. fabricated and analyzed the EC device. D.H.H. fabricated the sclera and face using a 3D printer. Y.J.C. and D.G.R. assisted with the conceptualization and system design. J.H.L. and J.H.P. performed the training/recognition task of MNIST digit patterns. J.W.C. and J.W.K. measured UPS and IPES to analyze the mid-gap trap. Y.Y.C. and S.B.J. assisted with the visualization and data analyses. All authors discussed the progress of the research and contributed to editing the paper.

## Competing interests

The authors declare no competing interests.
