## [Peer Review File · Nature Communications]

REVIEWER COMMENTS

Reviewer #1 (Remarks to the Author):

The authors have demonstrated the photonic synapse along with the system that mimics pupilar and corneal reflexes of the eye. The manuscript is well written, with a few missing parts. The novelty of this work is the integration of the photonic synapse with the electronic circuitry to build an ocular prosthesis system. I recommend this manuscript for publication after addressing the following comments.

1) The authors have shown a single QEPS which is a photonic synapse and integrated with the chemo-electro-mechanical system to mimic the functionality of the reflexes. In order to perform visual recognition, an array of photonic synapses are essential. How about the array of the QEPS devices? How to emulate reflexes in that case? This is not clear in the manuscript.

2) Page-5, Line-119, there is a discussion about figure 1f, but it is missing in the manuscript.

3) The authors have not discussed the derivation of the band diagram and values mentioned in Fig. 2D. The details have to be mentioned.

4) Page-6, 152-154, authors mentioned theoretical analysis. It is not clear what kind of theoretical analysis is followed to bring out the conclusion. There is no discussion about any simulations or references to support the discussion mentioned lines of the manuscript and supplementary information.

5) The wavelength of light is not mentioned in many experiments discussed in the manuscript. (Eg: Page-7, Line 186) It is to be mentioned.

6) The concept of removable charge is not clear in the manuscript. It requires more elaboration. If there is some removable charge in the case of QEPS devices, how are they reaching the initial state? In contrast with the PSC device. How is removable charge aiding devices to initial state after LTD?

7) How QEPS devices are reaching the initial state, even if there is some hysteresis in QEPS device (Supplementary Figure 5 and 6a)? But not in the case of IPS devices.

8) Define time constants used in supplementary Table-1. They are not clear to understand.

9) How about wavelength dependence on PEC/PO (Fig 3F) characteristics?

10) How about the variation PSC of QEPS devices for multiple cycles of Light ON and OFF?

11) Mention the dimensions of the components or the scalebar in Supplementary Fig-16.

Reviewer #2 (Remarks to the Author):

In this paper, the authors checked the photoinduced synaptic properties of the quantum dot embedded photonic synapse as a confirmatory test. Following this, they have emulated the pupillary light reflex by using the quantum dot embedded photonic synapse, electrochromic device, and CMOS components. In addition, they have connected a solenoid to the pupillary light reflex system to emulate the corneal reflex. QD embedded photonic synapses is not at all a new concept. It has been reported many times previously. This is same with respect to EC and Solenoid devices. The major claim in this paper is the synergistic function of electrochromic device, photonic synapse and solenoid. The only interesting part is the integration of all the devices.

1. What is the source of trap in InP QDs. Is it just the surface defect or due to the functional groups attached. How the trap density is controlled from sample to sample?

2. What is the trap depth in these QDs beyond just the schematics shown? Do you have any control on it? As the STP and LTP is dependent on this trap depth, how illumination time affect it as there can be de-trapping due to sample heating too?

3. It is given, "These PPC characteristics and the removable charge can be utilized for the synaptic weight adjustment of the photonic synapse." I am wondering how synaptic weight adjustment can be done since the trap heights are not tuned.

4. Why short term potentiation is not shown? This must be demonstrated to confirm the working of the photonic synapses.

5. Another important characteristic is the paired-pulse facilitation (PPF). It is strange that the authors skipped these important characteristics of synapses. The authors should demonstrate short term depression too. This is very important to demonstrate the working of their devices though the authors claim that they focus on continuous light illumination.

6. It can be seen from SI Fig 10 and 11 the LTP is only for a short period of time, < 120s. How this can be useful for any practical applications?

7. There is no novelty with respect to Solenoid and EC devices.

8. The authors should demonstrate some real neuromorphic application like face recognition using their devices.

REVIEWER COMMENTS

Reviewer #1 (Remarks to the Author):

The authors have demonstrated the photonic synapse along with the system that mimics pupilar and corneal reflexes of the eye. The manuscript is well written, with a few missing parts. The novelty of this work is the integration of the photonic synapse with the electronic circuitry to build an ocular prosthesis system. I recommend this manuscript for publication after addressing the following comments.

1) The authors have shown a single QEPS which is a photonic synapse and integrated with the chemo-electro-mechanical system to mimic the functionality of the reflexes. In order to perform visual recognition, an array of photonic synapses are essential. How about the array of the QEPS devices? How to emulate reflexes in that case? This is not clear in the manuscript.

Reply: Thank you for the reviewer's valuable comment. To fabricate an array of QEPS, we patterned the QD embedded IGZO using conventional photolithography and chemical etching (35-37 vol% hydrochloric acid in distilled water). Al with a thickness of 40 nm was then deposited to form source/drain electrodes, corresponding to postsynaptic/presynaptic terminals, respectively. By incorporating a control unit composed of a decoder, sensing amplifier, and QEPS array, the signal can be divided into reflex and MNIST. Previously this study was focused on mimicking the human ocular system, which emulates the signal transmission for the reflexes occurring in the eye. However, we additionally performed the visual recognition using a 6×6 array of QEPS which shows a little deviation in LTP/D behavior. To fabricate the 6×6 array of QEPS with high uniformity a sufficient concentration of NH₄SCN solution and time was used for sufficiently replacing the ligand on the QD surface. We added the related experimental result and sentence on the array of QEPS in the manuscript as follows

Page 8: To further validate the uniformity of the QEPS, LTP/D characteristics were measured on a 6×6 array of QEPS as shown in **Supplementary Fig. 21**. The training/recognition task of Modified National Institute of Standards and Technology (MNIST) digit patterns was performed using the 36 devices. In this simulation, the array of QEPS shows an average recognition accuracy of 91.7% (**Supplementary Fig. 22**).

Supplementary Figure 21. Film uniformity for 36 QEPS. a Optical image of a 6×6 QEPS array. **b** LTP/D behavior of each QEPS. **c** Summary of the synaptic properties for 36 QEPS.

Supplementary Figure 22. Training/recognition tasks and plot of recognition rate for MNIST digit patterns. **a** Schematic illustration of two-layer perceptron-based ANN. Each layer consists of 400, 200, and 10 neurons for the input layer, hidden layer, and output layer, respectively. **b** Recognition rate as a function of number of training epochs for 36 QEPS. Each epoch consists of 8,000 training numbers. **c** Maximum recognition rates for 36 QEPS with a root standard deviation (RSD) of 0.32 %.

2) Page-5, Line-119, there is a discussion about figure 1f, but it is missing in the manuscript.

Reply: Thank you for the reviewer's valuable comment. We changed figure 1f to figure 1e to correct the typo.

Page 5: A stronger incident light intensity with unacceptable optical power triggers an actuation voltage (V_{Act}) in the neuron circuits (corneal reflex, right panel of **Figure 1e**).

3) The authors have not discussed the derivation of the band diagram and values mentioned in Fig. 2D. The details have to be mentioned.

Reply: Thank you for the reviewer's valuable comment. The band diagram was drawn assuming the charge transport through IGZO domains is governed by the mobility edge model as the polycrystalline films commonly do, where the trap energy distribution of IGZO is described by the exponential function below the mobile states ($E=0$). In this case, the characteristic temperature of the trap derived from the Mark-Helfrich equation directly describes the characteristics energy of the exponentially distributed traps. Here IGZO is the n-type semiconductor, so the distribution is obtained near the conduction band. We added the related sentence in the manuscript as follows.

Page 6: Through linear fitting in **Region Y**, l was calculated to be 3.30 and 2.14, indicating that the T_c values in the IPS and QEPS are 999.13 K (0.085 eV) and 643.3 K (0.055 eV), respectively. Here, the energy distribution was calculated using the relation $E_c = kT_c$, where E_c represents the thermal energy at characteristic temperature.

Figure 2. d Energy band diagram of the trap distribution in photonic synapses.

4) Page-6, 152-154, authors mentioned theoretical analysis. It is not clear what kind of theoretical analysis is followed to bring out the conclusion. There is no discussion about any simulations or references to support the discussion mentioned lines of the manuscript and supplementary information.

Reply: Thank you for the reviewer's valuable comment. We employed the mobility edge model to describe the charge transport of our system, for which the current-voltage characteristics were analyzed through Mark-Helfrich equation to extract the subgap distribution of shallow traps. It should firstly be noted that the large light modulation is driven by the light gating effect from deeper trap states of the isolated InP QD domains, while the charge transport itself is governed by IGZO domains rather than long range hopping through midgap traps of InP QDs. The parameters for traps derived from the model are consistent with the high PSC of QEPS. To avoid any confusions of the potential readers, we have included the description of the theoretical model used for the analysis and clearly indicated the literatures we referred to support the discussions, as reviewer pointed out. Although the energy trap distribution is less than 0.1 eV, it can affect the current flow in the IGZO channel. As we responded to the comment 3) by the reviewer 1, QEPS shows narrower energy distribution. In addition, a lower trap density of QEPS ($1.60 \times 10^{16} \text{ cm}^{-3}$) compared to that of IPS ($2.18 \times 10^{17} \text{ cm}^{-3}$) is calculated using the Mark-Helfrich equation. Therefore, the enhancement of charge transport in QEPS can be rationalized (**Supplementary Fig. 4**). We added the related sentence in the manuscript as follows.

Page 6: Furthermore, the trap densities of IPS and QEPS were calculated to be 2.18×10^{17} and $1.60 \times 10^{16} \text{ cm}^{-3}$, respectively, using the Mark-Helfrich equation. **Figure 2D** shows the schematic band diagram of the model. The QEPS exhibits a shallow trap density-of-states (DOS) distribution compared to the IPS, despite the low conductivity of InP QD in the IGZO layer. In addition to the lower trap density, this trap distribution supports that the movement of carriers in the QEPS is less disturbed by traps present in the channel³⁵⁻³⁷ (see also **Supplementary Fig. 4**), by which the higher PSC of QEPS under illumination can be explained.

5) The wavelength of light is not mentioned in many experiments discussed in the manuscript. (Eg: Page-7, Line 186) It is to be mentioned.

Reply: As the reviewer commented, we use 520 nm of light to induce the PPC characteristics in IGZO and fill mid-gap traps in InP QDs. We added the wavelength of light in the manuscript as follows.

Page 7: Both photonic synapses were exposed to 520 nm of light with an intensity of $0.5 \mu\text{W}$ for 60 s and placed in dark conditions for 60 s while continuously applying the V_{Inh} pulse train. Both photonic synapses exhibit LTP characteristics in response to the photo stimulus.

Page 8: The effective modulation of light intensity by the EC device is shown in **Figure 3f** and **Supplementary Fig. 24**. To ensure that the experimental conditions were equivalent to those of the overall system, a laser beam (520 nm of wavelength) was passed through the EC device (P_{EC}) while supplying a current of 1 nA, similar to the initial current level of the PSC.

Page 9: First, when the s-eyelid is in the open-state, the EC device in the bleached state transmits the input light (520 nm of wavelength) with a slight decrease in power.

Page 11: The electrical properties of the QEPS were measured using a Keithley 4200A-SCS unit (Tektronix) and a 520 nm laser (MLL-III-520L, CNILaser) at room temperature under vacuum conditions.

6) The concept of removable charge is not clear in the manuscript. It requires more elaboration. If there is some removable charge in the case of QEPS devices, how are they reaching the initial state? In contrast with the PSC device. How is removable charge aiding devices to initial state after LTD?

Reply: Thank you for the reviewer’s valuable comment. In this study, we denote a photo-generated carrier that can be removed by applying positive V_{Inh} as a removable charge. The IGZO channel shows the PPC characteristics under illumination state. Thus, the photo-generated carriers remain in the channel after removing the light. However, the PPC characteristics can be eliminated by introducing InP QDs that facilitate recombination under the application of positive V_{Inh} . Here, we apply a positive V_{Inh} pulse train during the LTD process. In this manner, the increased PSC can easily reach the initial state in the case of QEPS devices, whereas it is difficult to reach the initial state for the IPS. We added the related sentence in the manuscript as follows.

Page 6: In the case of the QEPS, a large difference in ΔQ is observed between the illumination and re-dark states, whereas there is no notable difference for the IPS. Here, we define the difference in ΔQ as removable charge which means photogenerated carrier that can be easily removed.

Page 7: In brief, the mid-gap trap of InP QDs causes a removable charge that can easily return to its original state after applying V_{Inh} under off-state.

Page 7: Note that the different recombination processes in the IPS and QEPS originate from the absence or presence of mid-gap traps in InP QDs.

7) How QEPS devices are reaching the initial state, even if there is some hysteresis in QEPS device (Supplementary Figure 5 and 6a)? But not in the case of IPS devices.

Reply: As the reviewer commented, the difference in hysteresis between QEPS and IPS devices causes the different LTD behaviors. The clockwise direction of hysteresis means that the photo-generated carriers can be removed by applying positive V_{Inh} . Namely, the removable charge makes the hysteresis. Therefore, QEPS can return to the initial state due to hysteresis, whereas IPS cannot reach the initial state. We added the related sentence in the manuscript as follows.

Page 6: Moreover, the difference in ΔQ between the illumination and re-dark state indicates the charges that can be removed from the IGZO channel by applying V_{Inh} , as shown in **Figure 2f**. In the case of the QEPS, a large difference in ΔQ is observed between the illumination and re-dark states, whereas there is no notable difference for the IPS. Here, we define the difference in ΔQ as removable charge which means photogenerated carrier that can be easily removed. Note that the hysteresis behavior in **Supplementary Fig. 6** originates from the removable charge.

8) Define time constants used in supplementary Table-1. They are not clear to understand.

Reply: As the reviewer commented we define time constants in the supporting information as follows.

Table S1. Performance characteristics of EC device.

The maximum transmittance contrast (ΔT_{max}) was measured from Figure 3e. The response time, defined as the period to achieve a 90% change in ΔT_{max} , is extracted from the time-dependent transmittance profile during coloring ($\Delta t_{\text{c},90\%}$) and self-bleaching processes ($\Delta t_{\text{b,open},90\%}$). The coloration efficiency (η) of the EC platform is evaluated through the slope of **Supplementary Fig. 23**.

Applied voltage (V)	ΔT_{max} (%)	$\Delta t_{\text{c},90\%}$ (s)	$\Delta t_{\text{b,open},90\%}$ (s)	η (cm ² /C)
0.7	~ 96.2	~ 37	~ 85	~ 123.0
0.8	~ 98.5	~ 15	~ 114	~ 148.9
0.9	~ 99.8	~ 8	~ 120	~ 162.6

9) How about wavelength dependence on PEC/P0 (Fig 3F) characteristics?

Reply: As the reviewer commented, we measured the wavelength dependence on P_{EC}/P_0 , which is comparable

to the absorption properties of EC device. We added the related experimental result and sentence in the manuscript and supporting information as follows.

Page 8: The effective modulation of light intensity by the EC device is shown in **Figure 3f** and **Supplementary Fig. 24**.

Supplementary Figure 24. Plot of P_{EC}/P_0 versus current, showing the light intensity modulation capability of the EC device at various wavelengths.

10) How about the variation PSC of QEPS devices for multiple cycles of Light ON and OFF?

Reply: Thank you for the reviewer's valuable comment. We added the related experimental result and sentence in the manuscript and supporting information as follows.

Page 7: In contrast, the presence of abundant removable charge allows the increased PSC of the QEPS to return to the initial level during the multiple cycles of illumination and dark conditions (**Supplementary Fig. 15**).

Supplementary Figure 15. LTP/D behavior of the QEPS during the multiple cycles of illumination and dark conditions.

11) Mention the dimensions of the components or the scalebar in Supplementary Fig-16.

Reply: As the reviewer commented we added the scalebar in the supporting information as follows.

Supplementary Figure 26. Photographic image of the S-eyelid.

Reviewer #2 (Remarks to the Author):

In this paper, the authors checked the photoinduced synaptic properties of the quantum dot embedded photonic synapse as a confirmatory test. Following this, they have emulated the pupillary light reflex by using the quantum dot embedded photonic synapse, electrochromic device, and CMOS components. In addition, they have connected a solenoid to the pupillary light reflex system to emulate the corneal reflex. QD embedded photonic synapses is not at all a new concept. It has been reported many times previously. This is same with respect to EC and Solenoid devices. The major claim in this paper is the synergistic function of electrochromic device, photonic synapse and solenoid. The only interesting part is the integration of all the devices.

Thank you for the reviewer's valuable comment. We indeed agreed that the novelty of the component devices alone might not be sufficient to demonstrate the impact of this work. The most significant novelty of this manuscript, however, lies in the first design and demonstration of an autonomically light modulating system that emulates pupil and corneal reflexes, rather than in the novel demonstration or the optimization of each component device thereof. Furthermore, in the course of such demonstration, we put emphasis on the design principles of synergistically functioning integration systems and the corresponding selection criteria for the components.

As the reviewer mentioned, EC devices and solenoid devices themselves are not novel at all. However, those components were judiciously selected to meticulously emulate the essential biological traits of pupil corneal reflexes, which are the gradient light modulation and immediate light blocking capabilities, respectively. The EC device, firstly, was selected considering the gradient optical attenuation through the EC device in accordance with the linear increase of PSC. The solenoid device was chosen due to its instantaneous movement that can accommodate the need for emulating prompt reflex upon exposure to variable illumination conditions. On the other hand, there are additional valuable aspect in our design of photonic synapses on top of the selection and integration alone. As the reviewer would be well aware of, improving the LTD properties is one of the critical challenges in the metal oxide semiconductor-based photonic synapse due to its PPC characteristics. In our demonstration, we provide a new possibility of resolving such issue by harnessing the functionality of QD as a controlled electron releaser for metal oxides rather than as a commonly used light absorber. As a result, it was possible to overcome the chronic issues of metal oxide photonic synapses and to simultaneously achieve both excellent LTP from desirable PPC of metal oxides and LTD induced by embedded electron releaser QD.

Namely, the key achievements of our work can be summarized as (i) the first attempt to design an autonomic artificial light modulating system which emulates pupil and corneal reflexes and (ii) the suggestion of a new type of photonic synapse based on metal oxide semiconductor. The elaborations for the key aspects are listed below.

1. The first attempt to establish an autonomically light modulating system:

Artificial organs have emerged as promising replacements for human organs damaged by aging, injury, or disease. A representative perceptive prosthesis is an artificial eye that provides vision and is responsible for 80% of the information perception by humans. Recent approaches associated with artificial eyes have achieved satisfactory recognition of external light, but research on mimicking human ocular reflexes according to the intensity of external light has not yet been reported despite the fact that the light modulation response is imperative to perceive objects clearly by receiving the appropriate amount of light from the environment. Considering that human ocular reflexes, *i.e.*, pupillary light reflex and corneal reflex, are governed by a synaptic weight update induced by a light stimulus, therefore the adoption of photonic synapses with bio-similarity that can realize analog signal transmission is crucial for realizing artificial illuminance modulation responses.

Considering these aspects, in this manuscript, a biomimetic ocular prosthesis system that can mimic the human light modulation response was demonstrated by incorporating solenoid-based artificial eyelid, an electrochromic device, and QD embedded photonic synapse (QEPS)-based neuron circuits. As a result, filtering of the excess light was enabled by blocking and attenuating the light intensity through the s-eyelid and EC devices, respectively. On the basis of the findings of this study, the proposed biomimetic prosthesis system capable of both corneal reflex and pupillary light reflex provides new insights for research on biomimetic prosthesis systems.

2. Designing a new type of photonic synapse based on meatal oxide semiconductor:

To demonstrate the biomimetic prosthesis systems, the desirable artificial synapse should have the abilities in both light sensing and signal processing to decrease the circuit complexity and increase the integration density. In addition, photonic synapse requires high reliability, superior stability, and large-area processing for constructing artificial eyes.

In this context, many researchers have dedicated efforts to the realization of photonic synapses in which the synaptic weights are adjusted by external light stimuli. The use of metal oxide semiconductors is advantageous because the photoinduced generation of metastable oxygen vacancies in the metal oxide semiconductors induces persistent photocurrent (PPC) characteristics, leading to long-term potentiation (LTP) characteristics under pulsed light stimuli. Despite the excellent properties of photonic synapses, implementing long-term depression (LTD) remains a critical challenge.

To overcome the LTD properties, several studies generated traps in IGZO (Sci. Rep. 6, 31991, 2016) or introduced additional layers (e.g., alkylated graphene oxide, Adv. Funct. Mater. 28, 1804397, 2018). In this manuscript, for the first time, QD was introduced to improve the LTD properties. In contrast to other QD-based photonic synapses (Adv. Mater. 30, 1802883, 2018 & Nanoscale Adv. 3, 5046-5052, 2021), the main absorbent of photocurrent is indium-gallium-zinc oxide, not QDs. In this research, the mid-gap trap originating from the unpassivated surface of QDs can enhance the LTD properties of IGZO without commonly observed adversary decrements in optoelectronic properties in QD-absorber-based devices. We further investigate the role of the mid-gap trap using the mid-gap trap engineered (QDs with the almost entirely passivated surface) QEPS. From this point of view, our QEPS suggests a new type and mechanism of the metal oxide semiconductor-based photonic synapse with the enhanced LTD properties.

1. What is the source of trap in InP QDs. Is it just the surface defect or due to the functional groups attached. How the trap density is controlled from sample to sample?

Reply: Thank you for the reviewer's valuable comment. The origin of the mid-gap trap is the surface defect, preferably the under-coordinated P in InP QDs as the reviewer mentioned. Here we introduce the thiocyanate ligands to effectively remove the native ligands for facile carrier extraction. Because thiocyanate functional group cannot passivate P dangling bond on the InP surface, we expect the trap density originating from dangling P remain unchanged. (Chem. Mater. 33, 6885, 2021). We added the related experimental result and sentence in the manuscript and supporting information as follows.

Page 6: The removable charge in QEPS is a result of the mid-gap trap originating from the P dangling bond in the NH_4SCN passivated InP QDs, as evidenced by the trap emission in the wavelength region above 800 nm^{38,39}, as shown in **Supplementary Fig. 7**.

Page 8: To further validate the uniformity of the QEPS, LTP/D characteristics were measured on a 6×6 array of QEPS as shown in **Supplementary Fig. 21**.

Supplementary Figure 21. Film uniformity for 36 QEPS. a Optical image of a 6×6 QEPS array. **b** LTP/D behavior of each QEPS. **c** Summary of the synaptic properties for 36 QEPS.

2. What is the trap depth in these QDs beyond just the schematics shown? Do you have any control on it? As the STP and LTP is dependent on this trap depth, how illumination time affect it as there can be de-trapping due to sample heating too?

Reply: Thank you for the reviewer's valuable comment. We further conducted the UPS and IPES measurement to investigate the trap depth. Through the measurement, a distinguishable difference in trap state was observed near the conduction band and valence band of InP QDs. PL data shows that the energy of the mid-gap trap from the valence band is 1.55 eV. In short, the mid-gap trap is located around 0.785 eV below the conduction band edge (recombination of trapped electrons takes place considering the Urbach tail). While the trap depth (**Supplementary Fig. 8**) is considered fixed from the under-coordinated surface P in InP QDs, HF treatment on InP QD effectively removes the mid-gap traps. We indeed find fewer mid-gap traps in the HF-treated InP QDs, evidenced from UPS and IPES measurement. The decrease in the mid-gap trap is also observed in the PL measurement resulting in radiative recombination at 742 nm (**Supplementary Fig. 16**).

Furthermore, we investigate the synaptic properties of HF-treated InP QDs embedded photonic synapses to clarify the effect of the mid-gap trap. EPSC data do not show notable changes by controlling the amount of mid-gap trap. However, IPS data exhibit that the mid-gap trap facilitates the recombination process in IGZO. Therefore, increased synaptic weight does not return to its original state during the LTD process for photonic synapses with few mid-gap traps. Also, we measure the temperature of the device during light exposure. There was no significant temperature change for 2 hours (**Figure R1**). We added the related experimental results and sentences in the manuscript and supporting information as follows.

Figure R1. Temperature difference under light illumination

Page 7: To characterize the mid-gap trap state, ultraviolet photoelectron spectroscopy (UPS) and inverse photoemission spectroscopy (IPES) of InP QD was further investigated (**Supplementary Fig. 8**). The unoccupied density of states (DOSs) near the conduction band edge exhibit clear tail states compare to the occupied DOSs near the valence band edge. The position of distribution near the conduction band edge is located broadly at 0.5 to 1.49 eV above the fermi level (E_f). The tail states are associated with the trap states near the conduction band edge and the PL emission above 800nm. The energy difference between mid-gap trap and conduction band edge (~ 0.8 eV) is reduced due to the localized tail states (~ 0.25 eV) which facilitates the field-assisted detrapping to IGZO.

Page 7: To verify the role of the mid-gap trap in QEPS, we further investigate the mid-gap trap engineered QEPS through HF treatment^{40,41} (**Supplementary Fig. 16**). The result exhibits a similar tendency in the excitatory postsynaptic current (EPSC) whereas comparable change was observed in the inhibitory

postsynaptic current (IPSC) (**Supplementary Figs. 17 and 18**). As a result of the different IPSC behavior exhibited by the increase in the mid-gap trap, sufficient removable charge facilitates returning the increased PSC to its original state, as shown in **Supplementary Fig. 19**.

Supplementary Figure 8. DOSs distribution of InP QDs. a light absorption, **b** PL emission spectra, **c** UPS, and **d** IPES of HF-treated InP QDs. **e** Band structure of HF-treated InP QDs calculated from **a-d**.

Supplementary Figure 16. DOSs distribution of HF-treated InP QDs. a light absorption, **b** PL emission spectra, **c** UPS, and **d** IPES of HF-treated InP QDs. **e** Band structure of HF-treated InP QDs calculated from **a-d**.

To verify the mid-gap trap state, we treated the InP QDs using HF. During the treatment, P dangling bond on the surface of InP QDs is effectively passivated^{40,41}. Compared to the NH₄SCN-passivated InP QDs, HF-treated QDs show negligible mid-gap states near the conduction band edge which indicates that imperfect surface passivation occurs in the mid-gap trap in InP QDs. Due to the decrease in the mid-gap trap, radiative recombination is facilitated.

Supplementary Figure 17. EPSC behavior of the QEPS under application of various V_{inh} pulses. **a** 50mg/ml of InP QD solution was coated, **b** 25mg/ml of InP QD solution was coated, and **c** HF-treated InP QD solution was coated on the substrate, corresponding to high, intermediate, and low mid-gap trap density, respectively.

Supplementary Figure 18. IPSC behavior of the QEPS under application of various V_{inh} pulses. **a** 50mg/ml of InP QD solution was coated, **b** 25mg/ml of InP QD solution was coated, and **c** HF-treated InP QD solution was coated on the substrate, corresponding to high, intermediate, and low mid-gap trap density, respectively.

Supplementary Figure 19. LTP/D behavior of mid-gap trap engineered QEPS.

3. It is given, “These PPC characteristics and the removable charge can be utilized for the synaptic weight adjustment of the photonic synapse.” I am wondering how synaptic weight adjustment can be done since the trap heights are not tuned.

Reply: Thank you for the reviewer’s valuable comment. As reviewer mentioned, an increase in synaptic weight can be controlled through the PPC characteristic of IGZO due to the slow photo response time of IGZO. On the other hand, a synaptic weight can be reduced through removable charge which means the recombination of IGZO. In case of the recombination process in IPS, removing the photo generated carriers difficult due to the absence of the mid-gap trap. So, the presence of the mid-gap trap makes the removable charge which can adjust the synaptic weight. Increasing the trap height makes the de-trapping process more difficult, making it hard to return to its original state. On the other hand, decreasing the trap height makes the de-trapping process easier, further facilitating the recombination with poor nonlinearity.

4. Why short term potentiation is not shown? This must be demonstrated to confirm the working of the photonic synapses.

Reply: Thank you for pointing out the details we have missed. As the reviewer commented, we added the related experimental results and sentences in the manuscript and supporting information as follows.

Page 7: These PPC characteristics and the removable charge can be utilized for the synaptic weight adjustment of the photonic synapse (**Supplementary Figs. 11 and 12** show the short-term behavior of QEPS).

Supplementary Figure 12. Short-term behavior of the QEPS. a short-term potentiation (0.2 Hz and 0.1 Hz for left panel and right panel, respectively) and **b** short-term depression of the QEPS (0.2 Hz and 0.1 Hz for left panel and right panel, respectively).

5. Another important characteristic is the paired-pulse facilitation (PPF). It is strange that the authors skipped these important characteristics of synapses. The authors should demonstrate short term depression too. This is very important to demonstrate the working of their devices though the authors claim that they focus on continuous light illumination.

Reply: Thank you for pointing out the details we have missed. As the reviewer commented, we investigate PPF, PPD, and STD of QEPS and added the related experimental results and sentences in the manuscript and supporting information as follows.

Page 7: These PPC characteristics and the removable charge can be utilized for the synaptic weight adjustment of the photonic synapse (**Supplementary Figs. 11 and 12** show the short-term behavior of QEPS).

Supplementary Figure 11. Paired-pulse behavior of the QEPS. a PSC behavior and **b** paired-pulse facilitation (PPF) index of the QEPS. **c** PSC behavior and **d** paired-pulse depression (PPD) index of the QEPS.

Supplementary Figure 12. Short-term behavior of the QEPS. a short-term potentiation (0.2 Hz and 0.1 Hz for left panel and right panel, respectively) and **b** short-term depression of the QEPS (0.2 Hz and 0.1 Hz for left panel and right panel, respectively).

6. It can be seen from SI Fig 10 and 11 the LTP is only for a short period of time, < 120s. How this can be useful for any practical applications?

Reply: Thank you for the reviewer's valuable comment. To demonstrate the practical feasibility of our system, we measured LTP and LTD for a longer period (> 600 s) and multiple LTP/LTD cycles of QEPS. We added the related experimental result and sentence in the manuscript and supporting information as follows.

Page 7: In contrast, the presence of abundant removable charge allows the increased PSC of the QEPS to return to the initial level during the multiple cycles of illumination and dark conditions (**Supplementary Fig. 15**).

Supplementary Figure 15. LTP/D behavior of the QEPS during the multiple cycles of illumination and dark conditions.

7. There is no novelty with respect to Solenoid and EC devices.

Reply: We carefully selected solenoid and EC devices that can effectively modulate the input light, similar to pupil and corneal reflexes. Due to the gradual coloration behavior of the EC device during the linear increase in PSC and instantaneous movement of the solenoid device, the use of these components is suitable to fabricate the ocular system. The point of our study is the demonstration of a biomimetic ocular prosthesis system, and the most important part is the photonic synapse that can adjust the synaptic weights by external light stimuli. We designed the photonic synapse based on metal-oxide semiconductors to provide high stability. However, their PPC characteristics lead to poor LTD properties. Therefore, we introduced and engineered QDs to exploit mid-gap traps derived from their surface defects and improve LTD properties. This strategy is obviously different from conventional photonic synapses where QDs were employed for light absorption. From this point of view, our QEPS suggests a new type and mechanism of a photonic synapse.

8. The authors should demonstrate some real neuromorphic application like face recognition using their devices.

Reply: Thank you for pointing out the details we have missed. We added the related MNIST results and sentences in the manuscript and supporting information as follows.

Page 8: The training/recognition task of Modified National Institute of Standards and Technology (MNIST) digit patterns was performed using the 36 devices. In this simulation, the array of QEPS shows an average recognition accuracy of 91.7% (**Supplementary Fig. 22**).

Supplementary Figure 22. Training/recognition tasks and plot of recognition rate for MNIST digit patterns. **a** Schematic illustration of two-layer perceptron-based ANN. Each layer consists of 400, 200, and 10 neurons for the input layer, hidden layer, and output layer, respectively. **b** Recognition rate as a function of number of training epochs for 36 QEPS. Each epoch consists of 8,000 training numbers. **c** Maximum recognition rates for 36 QEPS with a root standard deviation (RSD) of 0.32 %.

REVIEWERS' COMMENTS

Reviewer #2 (Remarks to the Author):

The authors have adequately addressed my concerns. I recommend its acceptance.